# Inhibitory Effect of Cinnamaldehyde on Main Destructive Microorganisms of Nanhai No. 1 Shipwreck

Xinduo Huang [1], Yeqing Han [1], Jing Du [2], Peifeng Guo [1], Yu Wang [1], Kaixuan Ma [1], Naisheng Li [2], Zhiguo Zhang [2], Yue Li [3,*] and Jiao Pan [1,*]

[1] Key Laboratory of Molecular Microbiology and Technology of the Ministry of Education, Department of Microbiology, College of Life Sciences, Nankai University, Tianjin 300071, China; 2120191084@mail.nankai.edu.cn (X.H.); 2120191000@mail.nankai.edu.cn (Y.H.); Gary19981114@163.com (P.G.); 2120201036@mail.nankai.edu.cn (Y.W.); makaixuan16@163.com (K.M.)

[2] National Center of Archaeology, Beijing 100013, China; ldusts@163.com (J.D.); lineas@126.com (N.L.); zzgwys@126.com (Z.Z.)

[3] State Key Laboratory of Component-Based Chinese Medicine, Ministry of Education Key Laboratory of Pharmacology of Traditional Chinese Medicine Formulae, Institute of Traditional Chinese Medicine, Tianjin University of Traditional Chinese Medicine, Tianjin 301617, China

\* Correspondence: liyue2018@tjutcm.edu.cn (Y.L.); panjiaonk@nankai.edu.cn (J.P.)

**Abstract:** Nanhai No. 1, a shipwreck in the Southern Song Dynasty, China, has a history of more than 800 years. It was salvaged in 2007 and is now on display in the Guangdong Maritime Silk Road Museum. Due to the fact that the hull is a wooden cultural relic and exposed to the air, the biological corrosion and biodegradation caused by microorganisms are key problems of hull protection. At present, the antimicrobial agent Euxyl® K100 (isothiazolinone) has a significant antimicrobial effect in the field, but it has a certain negative impact on the environment and archeologists. In order to reduce the use of chemical antimicrobial agents, we evaluated the inhibitory effects of cinnamaldehyde on the main destructive microorganisms of Nanhai No. 1. Cinnamaldehyde is the main active component of cinnamon, and has broad-spectrum antimicrobial properties. The paper diffusion method, gas diffusion method and minimum inhibitory concentration experiment were used to detect the inhibitory effects of cinnamaldehyde on the main microorganisms of Nanhai No. 1. We found that cinnamaldehyde had significant inhibitory effects on *Bacillus tequilensis* NK-NH5, *Bacillus megaterium* NK-NH10, *Bacillus velezensis* NK-NH11, *Bacillus* sp. NK-NH15, *Bacillus* sp. NK-NH16, *Bacillus* sp. NK-NH17, *Fusarium solani* NK-NH1 and *Scedosporium apiospermum* NK.W1-3. At the same time, cinnamaldehyde had more inhibitory effects on fungi than bacteria. Finally, we verified that cinnamaldehyde can effectively inhibit the growth of microorganisms in water, for storing the scattered wood blocks of the Nanhai No. 1 hull through laboratory simulation experiments. Cinnamaldehyde, as an environment-friendly antimicrobial agent, is of great significance to protecting water-saturated wooden relics from microbial corrosion and degradation in the future.

**Keywords:** Nanhai No. 1 shipwreck; cinnamaldehyde; antimicrobial activity; biodegradation

## 1. Introduction

Nanhai No. 1, a Chinese Song Dynasty shipwreck, was discovered in 1987 near the Chuanshan archipelago in the South China Sea, Guangdong Province. In 2007, Nanhai No. 1 was salvaged and displayed in "the Crystal Palace" of the Guangdong Maritime Silk Road Museum. In 2012, archaeological excavation was officially started [1], and a large number of precious cultural relics such as gold, silver, bronze, iron, lacquer and porcelain were found in the wreck. The discovery of Nanhai No. 1 wreck is of great significance to Chinese underwater archaeology, and also provides important clues for the study of Maritime Silk Road.

The hull and the cultural relics of Nanhai No.1 are the key protected objects. With the excavation of Nanhai No.1, the exposed area of the hull increases gradually. The cultural

relics need to go through the process of desalination, desulfurization and iron removal, so they are soaked in a specific buffer for a long time [2]. Due to the high humidity and high temperature in the storage environment of Nanhai No. 1, and the fact that wooden cultural relics are natural organic matter, this provides suitable conditions for the growth and reproduction of microorganisms [3]. Microorganisms are very harmful to wooden relics. Some bacteria and fungi can destroy wood by degrading cellulose, hemicellulose and lignin. For example, the ability of microorganisms such as *Trichoderma viride*, *Trichoderma reesei*, *Bacteroides cellulosolvens*, *Bacteroides succinogenes* to degrade cellulose has been reported [4]. This will cause structural and mechanical changes of wooden cultural relics, and cause irreversible impact on precious cultural artifacts [5–7].

The main microorganisms of Nanhai No. 1 isolated by our previous research group include: *Fusarium solani* NK-NH1 [8], *Bacillus tequilensis* NK-NH5, *Bacillus megaterium* NK-NH10, *Bacillus velezensis* NK-NH11 [9] and *Scedosporium apiospermum* NK.W1-3 [10]. In addition, we have recently identified bacteria on the hull and surrounding sea mud including *Bacillus* sp. NK-NH15, *Bacillus* sp. NK-NH16, *Bacillus* sp. NK-NH17 (Table S1).

At present, Euxyl® K100 is the main antimicrobial agent used in Nanhai No.1 microbial control. Although it can effectively inhibit the growth of microorganisms, it is a little harmful to the environment and human body. Methylchloroisothiazolinone (MCI) and methylisothiazolinone (MI), the main components of Euxyl® K100, are widely used in cosmetic products [11]. MI and MCI can cause allergic contact dermatitis [12]. At the same time, MI has a certain volatility; it will slowly volatilize into the air, affecting the surrounding environment, and then cause harm to the contacts [13]. Therefore, it is an important task to find environment-friendly and health-friendly antimicrobial agents. Plant-derived antimicrobial agents are relatively safe and friendly to the human body and environment [14,15]. At present, there are many reports that plant-derived antimicrobial agents are used in the actual control of microbial diseases [16]. Our research goal is to apply plant-derived antimicrobial agents to the microbial control of the Nanhai No. 1 shipwreck.

Cinnamaldehyde is the main component of cinnamon essential oil, which is a yellow oily liquid [17]. The antiinflammatory, antioxidation, antiulcer, antibacterial, hypoglycemic and hypolipidemic properties of cinnamaldehyde have been reported [18]. Cinnamaldehyde has been widely used in food, medicine, cosmetics and other fields [19]. Previous studies have shown that cinnamaldehyde has good inhibitory effect on *Aspergillus*, *Fusarium*, *Penicillium*, *Rhizopus* and other fungi [20–22], *Escherichia coli*, *Bacillus subtilis*, *Staphylococcus* spp. and other bacteria [23,24]. In this study, we first tested the cellulose degradation ability of the main destructive microorganisms of Nanhai No. 1, then tested the inhibition effect of cinnamaldehyde on these main destructive microorganisms. In order to play a practical role in the microbial control of Nanhai No. 1, the simulated experiment was performed in the laboratory, and the inhibitory effect of cinnamaldehyde on microorganisms of the wood blocks of Naihai No.1 was evaluated.

## 2. Materials and Methods

### 2.1. Main Destructive Microorganisms of Nanhai No. 1 Shipwreck

In the previous work, we isolated and purified a variety of main destructive microorganisms from the samples of the Nanhai No.1 shipwreck. The details are as follows: *Bacillus tequilensis* NK-NH5, *Bacillus megaterium* NK-NH10, *Bacillus velezensis* NK-NH11 were isolated from the water samples of lacquerware plates [9]. *Bacillus* sp. NK-NH15, *Bacillus* sp. NK-NH16, and *Bacillus* sp. NK-NH17 were isolated from the hull wood and sea mud. *Fusarium solani* NK-NH1 is a fungus isolated from the wood of the shipwreck. *Scedosporium apiospermum* NK.W1-3 is a fungus isolated from water samples of wood storage [8,10]. All microorganisms were frozen at −80 °C.

### 2.2. Revitalizing the Microorganisms

The strain was taken out from the refrigerator ($-80\ °C$) and inoculated in Luria Broth (LB) agar medium. After one day's culture at 37 °C, a single colony was selected and inoculated in the new LB agar medium again. The strain was cultured at 37 °C for one day for subsequent experiments; the fungi were inoculated in potato dextrose agar (PDA) medium and cultured at 28 °C for 3 days, then a single colony was selected and inoculated in the new PDA medium again and cultured at 28 °C for 3 days for subsequent experiments.

### 2.3. Determination of Cellulase Activity in Destructive Microorganisms

Two types of carboxylmethylcellulose (CMC) agar media were prepared to evaluate the ability of the microorganisms to degrade cellulose: (1) CMC agar media for bacteria: CMC Na 15.0 g, NaCl 5.0 g, $KH_2PO_4$ 1.0 g, $MgSO_4$ 0.2 g, peptone 10.0 g, yeast extract 5.0 g, agar 18.0 g, and distilled water 1 L; (2) CMC agar media for fungi: $NaNO_3$ 2 g, $K_2HPO_4$ 1 g, $MgSO_4$ 0.5 g, KCl 0.5 g, CMC Na 2 g, peptone 2 g, agar 17 g and 1 L distilled water. The bacteria were put into the center of the plate, cultured at 28 °C for 4 days, and then the plate was dyed with 1 g/L Congo red solution. After 15 min, the dye was discarded, then 1 mol/L NaCl solution was added for washing, and 15 min later, the sodium chloride solution was discarded and the colony diameter and transparent circle diameter were determined [25]; a fungal disk with a diameter of 7.5 mm was cut from the edge of the active colony. Then, the disk was transferred to a CMC agar plate, cultured at 28 °C for 4 days, then 5 mL iodine-potassium iodide solution (2.0 g potassium iodide, 1.0 g iodine, 300 mL double distilled water) was added, and incubated in the dark at room temperature for 5 min to determine the colony diameter and transparent circle diameter [26]. According to the ratio of Hyaline circles (H) to colony diameter (D), the cellulase activity of the tested bacteria can be determined preliminarily. The higher the H/D value, the stronger the ability of cellulose degradation. All media were autoclaved at 121 °C for 20 min.

### 2.4. Preparation of Bacterial Culture Suspension and Spore Suspension of Fungi

The bacteria in LB agar medium were inoculated into LB liquid medium and cultured overnight at 180 rpm/min and then the bacterial standard curve was prepared. The final concentration of the bacterial solution was diluted to $10^6$ CFU/mL. Then, 5 mL 1/1000 Tween-80 solution was added into the purified single colony plate (fungi), the fungal spores were scraped with a glass rod, the spore solution was filtered out through sterile gauze, the number of spores was detected with a blood cell counting plate, and the final concentration of spore suspension was diluted to $1 \times 10^4$ CFU/mL. This suspension was used in the subsequent antimicrobial experiment.

### 2.5. Disk Diffusion Experiment

The inhibition of cinnamaldehyde (Rhawn, Shanghai, China) was determined by disk diffusion method. Cinnamaldehyde was dissolved in the mixture of 10/1000 dimethyl sulfoxide (DMSO) and 1/1000 Tween-80 and mixed evenly to make the final concentration of cinnamaldehyde 100 mg/mL and 50 mg/mL respectively. 0.5% Euxyl® K100 (Schülke, Norderstedt, Germany) was used as the positive control group, and the mixed solution of 10/1000 DMSO and 1/1000 Tween-80 was used as the negative control group. For bacteria, 100 μL bacterial culture solution was added to the LB agar plate and then spread evenly with a glass spreading rod. Four pieces of sterile filter paper with diameters of 7 mm were placed on each plate, then 15 μL of 50 mg/mL cinnamaldehyde, 100 mg/mL cinnamaldehyde, 0.5% Euxyl® K100, mixed solution of 10/1000 DMSO and 1/1000 Tween-80 were added on the corresponding filter paper sheets. After incubation at 37 °C for 24 h, the size of inhibition zone was observed. The larger the diameter of the inhibition zone was, the stronger the inhibitory effect of the drug on bacteria was. For fungi, 100 μL spore suspension was added to a PDA plate and evenly coated with a sterile cotton swab. A piece of filter paper was placed on each plate and 15 μL drug or mixed solution was dropped.

After being cultured at 28 °C for 3 days, the size of the inhibition zone was observed. All materials (media, cotton swabs, etc.) were autoclaved at 121 °C for 20 min.

### 2.6. Antimicrobial Experiment of Cinnamaldehyde Volatile Gas

The antimicrobial effect of cinnamaldehyde volatile gas was determined by dichotomy plate. LB agar medium (bacteria) or PDA medium (fungi) was poured into one side of a split plate. After the medium solidified, 100 μL bacterial culture suspension or spore suspension of fungi were taken and coated evenly with a sterile cotton swab on one side of the plate. The air concentration of cinnamaldehyde was 250 μL/L when 20 μL cinnamaldehyde was added to the other side and the volume of the 90 mm culture dish was about 80 mL. After 24 h at 37 °C, the inhibition area was observed and the inhibition rate was calculated. Mixed solvents were used as negative control, and Euxyl® K100 as positive control.

$$\text{Inhibition rate} = \text{inhibition area}/(\text{colony area} + \text{inhibition area}) \times 100\%$$

### 2.7. Determination of Minimum Inhibitory Concentration of Cinnamaldehyde

Minimum inhibitory concentration assays were performed in 96 well plates. Cinnamaldehyde and Euxyl® K100 were diluted by the double dilution method using a mixed solution of 10/1000 (DMSO) and 1/1000 Tween 80. The concentrations of cinnamaldehyde ranged from 0.003125 mg/mL to 2 mg/mL and the concentrations of Euxyl® K100 ranged from 0.003906 mg/mL to 5 mg/mL. Different concentrations of drugs (50 μL) and bacterial culture suspension or spore suspension of fungi (150 μL) were added to each well. Then, 150 μL bacterial culture suspension/spore suspension and 50 μL mixed with solution of 20/1000 DMSO and 1/1000 Tween-80 were added to the wells as negative control, Euxyl® K100 as positive control, 150 μL medium and 50 μL mixed solution of 10/1000 DMSO and 1/1000 Tween-80 were added to the wells as blank control. The 96 well plates with fungi were kept at a constant temperature of 28 °C for 36 h, and the 96 well plates with bacteria were kept at a constant temperature of 37 °C for 24 h. At the end of incubation, the growth of microorganisms was determined by measuring the absorbance of each well at 595 nm (fungi)/600 nm (bacteria) with a multifunctional microplate reader, calculating the minimum inhibitory concentration, and each test was done in three replicates.

### 2.8. Laboratory Simulation Experiments

Five pieces of Nanhai No. 1 hull wood samples (length: 3 cm; width: 2 cm; height: 1 cm) were taken and placed in five boxes with volumes of 5 L. Two liters of distilled water was added into each box, and 500 μL of each bacterial culture suspension (*Bacillus* sp. (NK-NH15), *Bacillus* sp. (NK-NH16), *Bacillus* sp. (NK-NH17), *B. megaterium* (NK-NH10), *B. velezensis* (NK-NH11), *B. tequilensis* (NK-NH5) was inoculated in the boxes. Five groups (N, C1, C2, C3, C4) were set up. Cinnamaldehyde (0 g, 0.1 g, 0.4 g, 1 g, 2.5 g) was added sequentially to ensure the final concentration of cinnamaldehyde in the solution was 0, 50, 200, 500, 1250 μg/mL, respectively; after stirring evenly, boxes were sealed with a lid and allowed to stand for 14 days at an ambient temperature of 18–25 °C. The annual temperatures in the Marine Silk Road Museum are between 16.4–30.7 °C.

### 2.9. Determination of Colony Forming Units(CFU)

At the end of the laboratory simulation experiment, the water samples were stirred well with a glass rod, and 1 mL of water sample was taken out of each box for the determination of the total amount of bacteria. Water samples from each group were subjected to ten-fold gradient dilution to give a final concentration of $10^0$–$10^{-3}$. 100 μL of water samples at each concentration were taken and dropped onto LB agar plates, respectively, and spread well with a sterile coating rod before being incubated for 24 h. Colony forming units were counted. Each concentration was repeated three times.

### 2.10. Quantitative Real-Time PCR

After the laboratory simulation experiment, the water samples in each box were passed through a suction filter device for bacteria collection, the microorganisms were cut off on a 0.22 μM filter membrane, and then DNeasy powersoil Kit (Qiagen, Germany) was employed to extract the total DNA in the water samples. The total biomass of bacteria was determined by qPCR using primers Eub338(5′-ACTCCTACGGGAGGCAGCAG-3′)/Eub518(5′-ATTACCGCGGCTGCTGG-3′) targeted on the 16S rRNA gene [27]. The standard curve was constructed by drawing the logarithm of the concentration of seven gradient dilutions of genomic DNA and the threshold period (Ct value) produced by qPCR analysis. *E. coli* genomic DNA was used as the standard template for quantitative detection of bacteria. QPCR was performed in a Step One Plus ™ real-time PCR system using faststart universal SYBR Green master (Rox) (Roche, Switzerland). Each PCR reaction included 1 μL DNA template, 2 μL 10 mM bacterial primers Eub338 / Eub518, 10 μL SYBR green mix, and 7 μL $H_2O$ for a total of 20 μL. The cycling program consisted of denaturation at 95 °C for 10 min, followed by 40 cycles of 95 °C for 15 s, 54 °C for 30 s, 72 °C for 20 s. Melting curve analysis was established by increasing the temperature from 60 to 95 °C.

## 3. Results

### 3.1. Cellulolytic Enzyme Activities of Hull Main Microorganisms

To examine the extent of microbial influence on the hull, we determined the cellulose degrading capacity of these destructive microbes. As shown in Figure 1, all of these strains were able to degrade cellulose to generate degradation circles (H/D: 1.1–2.8), except *B. megaterium* (NK-NH10) which could not. Comparing the cellulose degradation ability of several strains of bacteria, *Bacillus* sp. (NK-NH17), *B. tequilensis* (NK-NH5) showed the highest degradation ability (H/D > 2.5), *Bacillus* sp. (NK-NH 15), *Bacillus* sp. (NK-NH16), and *B. velezensis* (NK-NH11) also showed a high capacity for cellulose degradation (H/D = 1.5–2.5). *F. solani* (NK-NH1) and *S. apiospermum* (NK. W1-3) had weak degradation ability (H/D < 1.5). These destructive microorganisms, by degrading cellulose of the hull or other organic matter cultural relics, can cause irreversible damage to these cultural relics. The control of microorganisms is important for the conservation of Nanhai No. 1.

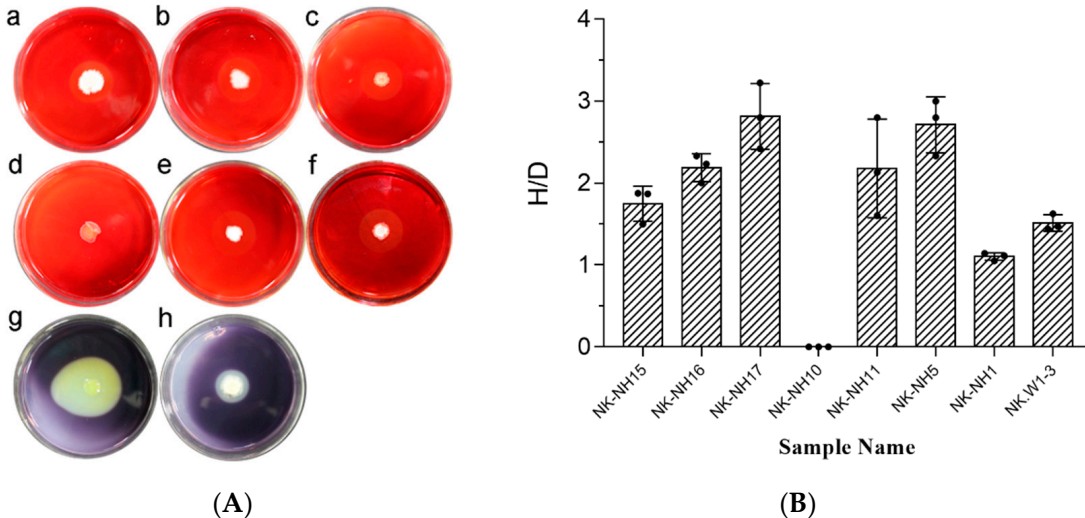

(**A**)  (**B**)

**Figure 1.** Cellulase activity of the main destructive microorganisms in Nanhai No. 1. (**A**) Hyaline circles and colony growth on CMC plates of the strains to be tested. (**a**) *Bacillus* sp. (NK-NH15), (**b**) *Bacillus* sp. (NK-NH16), (**c**) *Bacillus* sp. (NK-NH17), (**d**) *B. megaterium*(NK-NH10), (**e**) *B. velezensis* (NK-NH11), (**f**) *B. tequilensis* (NK-NH5), (**g**) *F. solani* (NK-NH1), (**h**) *S. apiospermum* (NK.W1-3). (**B**) The diameter of the colonies and the surrounding hyaline circles on the CMC plates were determined, denoted by D and H, respectively. The ratio of H and D (H/D) can give a preliminary indication of the cellulase activity of each colony.

### 3.2. Determination of Antimicrobial Activity of Cinnamaldehyde by Disc Diffusion Method

To determine the inhibitory activity of cinnamaldehyde against the tested microorganisms, we first employed the disk diffusion method. As shown in Figure 2, different sizes of inhibition zones were all produced around the filter paper sheets containing cinnamaldehyde, which illustrated that, at a certain concentration, cinnamaldehyde had inhibitory activities on the strains to be tested. As the concentration of cinnamaldehyde increased, the more obvious of an inhibitory effect was observed. Cinnamaldehyde at the concentration of 100 mg/mL showed strong inhibitory ability against *Bacillus* sp. (NK-NH15), *Bacillus* sp. (NK-NH16), *Bacillus* sp. (NK-NH17), *B. megaterium*(NK-NH10), *B. velezensis* (NK-NH11), *B. tequilensis* (NK-NH5), *F. solani* (NK-NH1), and *S. apiospermum* (NK.W1-3), and the diameter of each inhibition zone was 2.01, 2.51, 2.08, 1.89, 2.34, 2.55, 4.77, and 9 cm, respectively. The antimicrobial effect of 100 mg/mL cinnamaldehyde was similar to that of 0.5% Euxyl® K100. The inhibitory ability of cinnamaldehyde on fungi was stronger than that of bacteria. We found that *S. apiospermum* (NK.W1-3) was very sensitive to cinnamaldehyde. Cinnamaldehyde of 50 mg/mL could completely inhibit the growth of *S. apiospermum* (NK.W1-3), but 0.5% Euxyl® K100 could not.

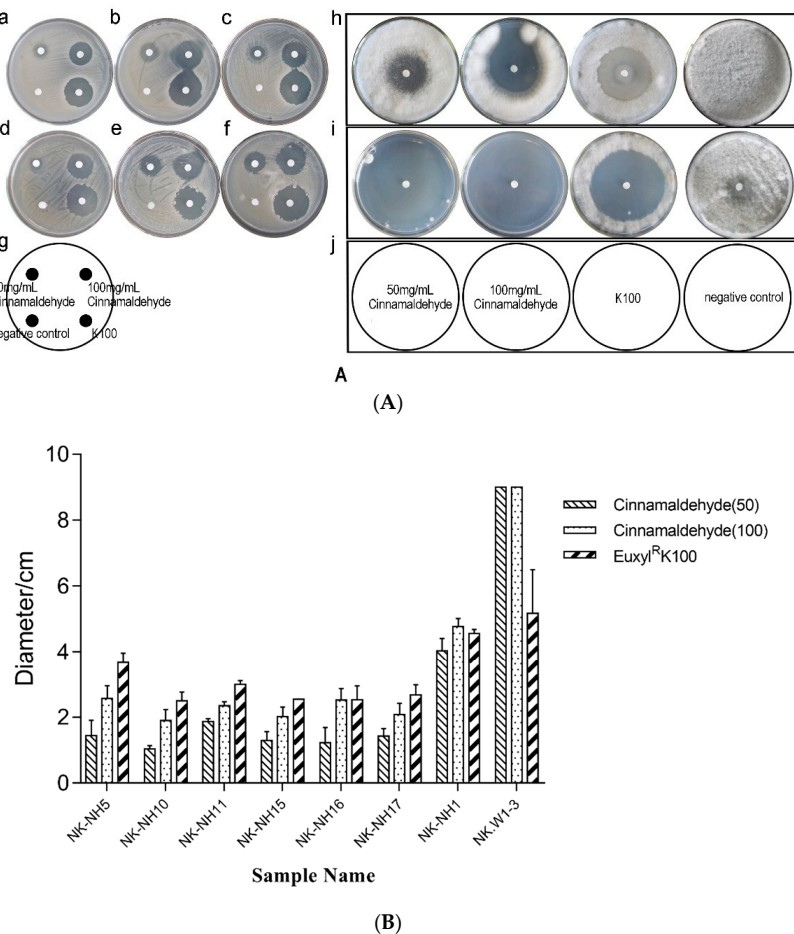

**Figure 2.** Inhibitory activities of cinnamaldehyde against the major destructive microorganisms of Nanhai No. 1. (**A**) Drugs created a zone of inhibition for different strains. (**a**) *Bacillus* sp. (NK-NH15), (**b**) *Bacillus* sp. (NK-NH16), (**c**) *Bacillus* sp. (NK-NH17), (**d**) *B. megaterium*(NK-NH10), (**e**) *B. velezensis* (NK-NH11), (**f**) *B. tequilensis* (NK-NH5), (**g**,**j**) drug distribution, (**h**) *F. solani* (NK-NH1), and (**i**) *S. apiospermum* (NK.W1-3); (**B**) cinnamaldehyde (50), cinnamaldehyde at concentration of 50 mg/mL; cinnamaldehyde (100), cinnamaldehyde at concentration of 100 mg/mL; Euxyl® K100, the concentration of Euxyl® K100 was 0.5%.

### 3.3. Determination of Antimicrobial Activity of Cinnamaldehyde Volatile Gas in Air

To determine the inhibitory activity of cinnamaldehyde volatile gas in air against destructive microorganisms, we completed the experiment using dichotomy plates. As shown in Figure 3, when the volatile gas concentration of cinnamaldehyde is 250 μL/L, cinnamaldehyde exerted antimicrobial activity on all the tested strains, while Euxyl® K100 did not (Figure S1). Specifically, cinnamaldehyde exhibited the strongest inhibitory effect against *S. apiospermum* (NK.W1-3) with an inhibition rate of 100%, and against *F. solani* (NK-NH1) with an inhibition rate of 45%.

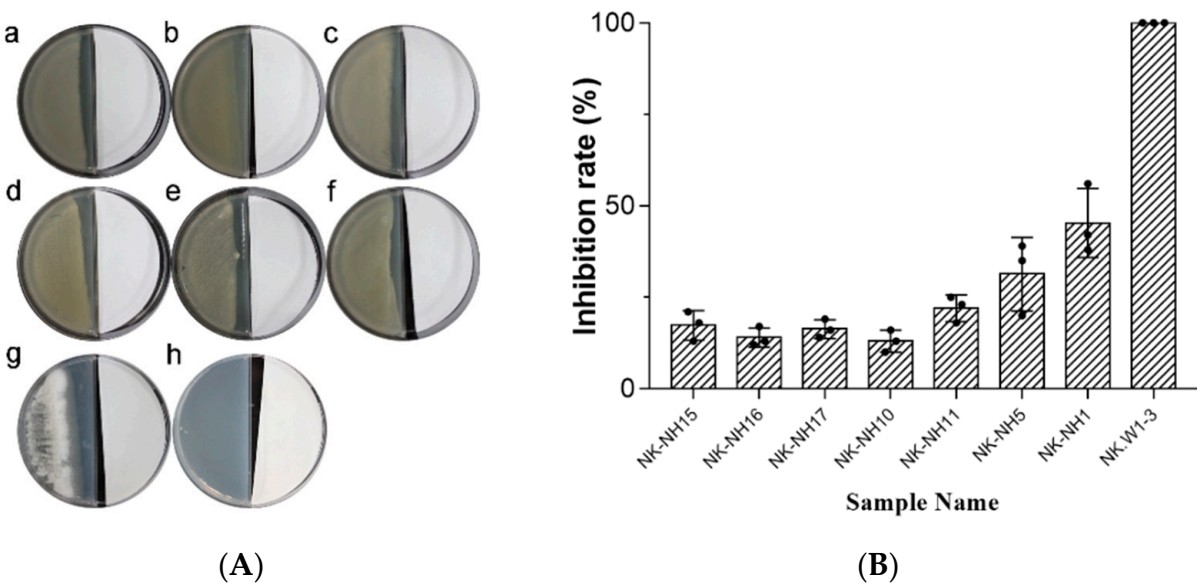

**(A)**                                                 **(B)**

**Figure 3.** Antimicrobial activity of cinnamaldehyde volatile gas in air (**A**) microbial growth on the dichotomy plate; left (bacteria), right (cinnamaldehyde 250 ul/L). (**a**) *Bacillus* sp. (NK-NH15), (**b**) *Bacillus* sp. (NK-NH16), (**c**) *Bacillus* sp. (NK-NH17), (**d**) *B. megaterium*(NK-NH10), (**e**) *B. velezensis* (NK-NH11), (**f**) *B. tequilensis* (NK-NH5), (**g**) *F. solani* (NK-NH1), and (**h**) *S. apiospermum* (NK.W1-3). (**B**) Inhibition rates of cinnamaldehyde against eight microorganisms to be tested.

Similar to the results of the disk diffusion method, the inhibitory ability of cinnamaldehyde volatile gas to bacteria is relatively weak, and for *Bacillus* sp.(NK-NH15), *Bacillus* sp.(NK-NH16), *Bacillus* sp.(NK-NH17), *B. megaterium*(NK-NH10), *B. velezensis* (NK-NH11), and *B. tequilensis* (NK-NH5), the inhibition rate of bacteria is 17%, 14%, 17%, 13%, 22%, and 31%, respectively.

### 3.4. MIC Determination of Cinnamaldehyde

To compare the inhibitory ability of cinnamaldehyde, the minimum inhibitory concentration of eight destructive microorganisms was measured. As shown in Table 1, cinnamaldehyde for *Bacillus* sp. (NK-NH15), *Bacillus* sp. (NK-NH16), *Bacillus* sp. (NK-NH17), *B. megaterium* (NK-NH10), *B. velezensis* (NK-NH11), *B. tequilensis* (NK-NH5) had inhibitory activity with respective MIC values of 1 mg/mL, 2 mg/mL, 0.125 mg/mL, 0.5 mg/mL and 1 mg/mL. Inhibition of Euxyl® K100 was somewhat more effective. *F. solani* (NK-NH1) and *S. apiospermum* (NK.W1-3) were more sensitive to cinnamaldehyde with MICs of 0.250, 0.0625 mg/mL, respectively. *S. apiospermum* (NK.W1-3) was more sensitive to cinnamaldehyde than Euxyl® K100 (0.07813 mg/mL).

**Table 1.** MIC values of cinnamaldehyde against destructive microorganisms of Nanhai No.1 (mg/mL).

| Strains | Cinnamaldehyde | Euxyl® K100 |
|---|---|---|
| *Bacillus* sp. (NK-NH15) | 1 | 0.07813 |
| *Bacillus* sp. (NK-NH16) | 2 | 0.03906 |
| *Bacillus* sp. (NK-NH17) | 1 | 0.03906 |
| *B. megaterium* (NK-NH10) | 0.125 | 0.07813 |
| *B. velezensis* (NK-NH11) | 0.5 | 0.03906 |
| *B. tequilensis* (NK-NH5) | 1 | 0.03906 |
| *F.solani* (NK-NH1) | 0.25 | 0.1563 |
| *S. apiospermum* (NK.W1-3) | 0.0625 | 0.07813 |

*3.5. Cinnamaldehyde Inhibitory of Major Destructive Microbes of Hull during Laboratory Simulation Experiment*

To verify the inhibitory effect exerted by cinnamaldehyde in practical applications, we simulated the protection status of hull wood from Nanhai No. 1. Shipwreck hull scattered wood was put in deionized water for moisture stabilization treatment and preliminary desalination treatment in the tank. Bacterial suspension of *Bacillus* sp. (NK-NH15), *Bacillus* sp. (NK-NH16), *Bacillus* sp. (NK-NH17), *B. megaterium* (NK-NH10), *B. velezensis* (NK-NH11) and *B. tequilensis* (NK-NH5) was added in the water, then covered and the contact of wood with air was reduced (Figure S2). After standing for 14 days, the total number of bacteria in the water samples was determined. As shown in Table 2, the total number of bacteria in water samples without cinnamaldehyde was $4.27 \times 10^5 \pm 2.04 \times 10^4$ CFU/mL. Cinnamaldehyde could inhibit microbial growth in water, and as the concentration of cinnamaldehyde rose, the number of microorganisms in the water continuously decreased. When the cinnamaldehyde concentration reached the highest value of 1.25 mg/mL, the total number of colonies in the water was $1.2 \times 10^1 \pm 5.10$ CFU/mL.

**Table 2.** Inhibitory effects of cinnamaldehyde on bacteria in water samples containing shipwreck wood.

| | N | C1 | C2 | C3 | C4 |
|---|---|---|---|---|---|
| CFU/mL | $4.27 \times 10^5 \pm 2.04 \times 10^4$ | $1.67 \times 10^5 \pm 9.02 \times 10^3$ | $1.2 \times 10^4 \pm 5.89 \times 10^2$ | $5.07 \times 10^2 \pm 3.86$ | $1.2 \times 10^1 \pm 5.10$ |

At the same time, the method of qPCR was applied to determine the biomass of bacteria in water samples. The correlation coefficient of the standard curve for *E. coli* was >0.99. (Figure 4A). As shown in Figure 4B, the biomass of bacteria showed a decreasing trend with increasing concentrations of cinnamaldehyde. When the concentration of cinnamaldehyde was 1.25 mg/mL, the concentration of bacterial DNA was $1.34 \pm 0.09$ ng/μL, which was significantly lower than that of the negative control ($584.65 \pm 52.26$ ng/μL).

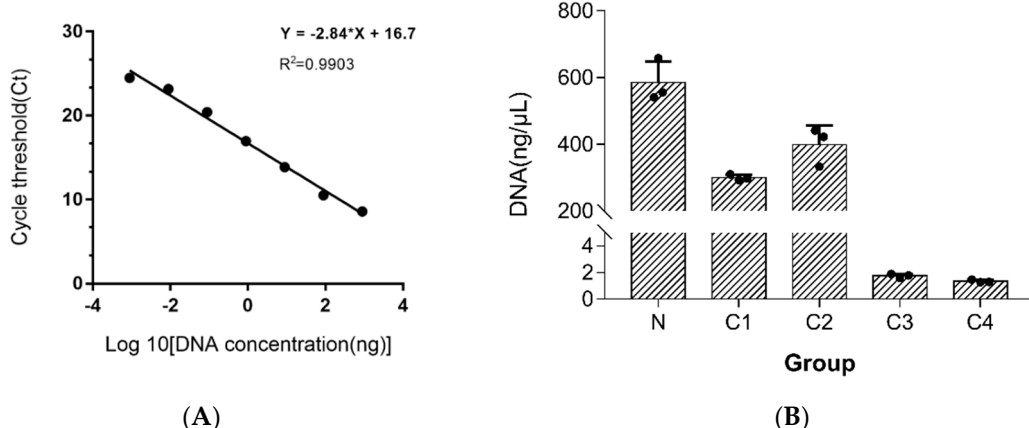

**Figure 4.** Quantification of bacterial biomass of samples by qPCR. (**A**) Standard curve for *E. coli* quantification using bacterial primers Eub338/Eub518. (**B**) Comparison of bacterial amounts. N: Water samples without cinnamaldehyde addition; C1: cinnamaldehyde concentration at 50 µg/mL; C2: cinnamaldehyde concentration at 200 µg/mL; C3: cinnamaldehyde concentration at 500 µg/mL; and C4: cinnamaldehyde concentration at 1250 µg/mL.

## 4. Discussion

The present archaeological digging work of Nanhai No. 1 is close to its end, with the hull chronically exposed to air and with high hull humidity (74.3%), which provides a suitable growth environment for microbial growth. Microorganisms, on the one hand, can affect the aesthetics of cultural relics, and on the other hand can cause harm to the cultural relics themselves. So, in the process of cultural relic protection, the prevention and control of microorganisms is of great significance. Previous studies have found that Nanhai No. 1 is affected by multiple microorganisms. *F. solani* (NK-NH1) is the major destructive fungus on ship wood, accounting for 90% of the total fungal load on the hull, which presents white plaques [8]. *S. apiospermum* (NK. W1-3) is a major destructive fungus in water samples stored in shipwreck wood. The genus *Scedosporium* accounted for 72.28% of the total fungi in water samples. *B. tequilensis* (NK-NH5), *B. velezensis* (NK-NH11), *B. megaterium* (NK-NH10) were the main bacteria in water samples storing wooden lacquer plates, *Bacillus* sp. (NK-NH15), *Bacillus* sp. (NK-NH16), *Bacillus* sp. (NK-NH17) were the main harmful bacteria isolated on ship wood. These destructive microorganisms can degrade cellulose and lignin, causing irreversible effects on hulls. Currently, the main measures for controlling microbial degradation of Nanhai No. 1 are to moisturize and inhibit the hulls by spraying them with distilled water containing 0.5% Euxyl® K100. Although the process can be effective in controlling microbial degradation of the hull, once the use of bacteriostatic agents is discontinued, the microorganism undergoes substantial reproductive growth. At the same time, Euxyl® K100 was found to cause some negative effects on the archaeological and relic protection workers' bodies. As such, finding safer and more environmentally friendly ways to inhibit these destructive microorganisms is very important. The aim of this study was to investigate the inhibitory activity of cinnamaldehyde against the destructive microorganisms of Nanhai No. 1. We first show that these destructive microorganisms are potentially harmful to hulls. Cellulose degradation experiments found that except for *B. velezensis* (NK-NH11), other microorganisms to be tested had the ability to degrade cellulose. The prevention and treatment of destructive microorganisms of Nanhai No. 1 should take both fungi and bacteria into account. Meanwhile we determined the antimicrobial activity of cinnamaldehyde. Previous studies have reported that cinnamaldehyde has a broad spectrum of antimicrobial activity [17,19,28,29]. We first preliminarily measured the inhibition of destructive microorganisms of Nanhai No. 1 by the disk diffusion method and found that cinnamaldehyde, at a certain concentration, could effectively inhibit microbial growth, which was comparable to the effects of Euxyl® K100 at 0.5%. At the same time, we found that fungi were more sensitive to cinnamaldehyde. Taking into

account the volatile properties of cinnamaldehyde, we measured the inhibition effect of cinnamaldehyde volatile gas in air. Similarly to the paper disk diffusion method results, airborne cinnamaldehyde can effectively inhibit microbial growth, although the inhibition effect is more obvious on fungi. Cinnamaldehyde, through its volatile properties, exerts antimicrobial ability, which can be used on the inhibition of destructive microorganisms of cultural relics in practical work. This provided us with ideas for the future practical application of cinnamaldehyde in the protection of cultural relics, and cinnamaldehyde can be slowly released in a closed vessel to achieve the effect of sustained antimicrobial activity. Edible films containing cinnamaldehyde have been reported to be effective in inhibiting *Salmonella* on vegetables sealed plastic bags [30]. In the results of MIC, cinnamaldehyde showed obvious antimicrobial activity, but lower than Euxyl® K100. For the antimicrobial mechanism of cinnamaldehyde has been intensively studied, and it has been reported that cinnamaldehyde can cause changes in the fatty acids of cell membranes and alter its structure, thus prompting the binding of cinnamaldehyde or other compounds to cells [31]. It also provides ideas for our later experiments: through the combined use of multiple antimicrobial agents, so as to achieve the effect of reducing the hazard and improving the drug efficacy.

The antimicrobial activity of cinnamaldehyde has been largely evaluated in laboratory experiments, while it is also widely used in the food industry [32]. The ability of antimicrobial agents against microorganisms is closely related to the host, the environment, and so on. To evaluate the feasibility of cinnamaldehyde application for microbial degradation control in Nanhai No. 1, we simulated the protection status of the scattered wood of Nanhai No. 1 and added Cinnamaldehyde to observe its bacteriostatic effect. The results showed that cinnamaldehyde could inhibit the growth of bacteria in water, and the concentration of bacteria in water decreased as the concentration of cinnamaldehyde increased. When the concentration of cinnamaldehyde is 0.5 mg/mL, it can inhibit the reproduction of most bacteria. The application of cinnamaldehyde to the protection of wooden cultural relics in water will provide a new way to develop environmentally friendly, safe and inexpensive bacteriostatic agents of plant origin in the future. In the following work, we will pay more attention to the practical use and effect of cinnamaldehyde in field protection, and the combined use of cinnamaldehyde with other antimicrobial agents is also of interest. In this study, cinnamaldehyde was found to inhibit the main destructive fungi of Nanhai No. 1, which is comparable to the Euxyl® K100 effect used for conservation on-site and also has some inhibitory activity against bacteria. Cinnamaldehyde not only showed antimicrobial ability in laboratory experiments, but also effectively inhibited the reproductive growth of bacteria in simulated experiments, indicating that cinnamaldehyde has a promising application in the prevention and control of cultural relic biodegradation.

**Supplementary Materials:** The following are available online at https://www.mdpi.com/article/10.3390/app11115262/s1, Figure S1: Microbial growth on the dichotomy plate, Figure S2: The state of shipwreck hull scattered wood in water tank. Table S1: Molecular identification of strains isolated from the hull, sea mud.

**Author Contributions:** Conceptualization, J.P. and Y.L.; methodology, X.H. and Y.H.; software, X.H. and Y.H.; validation, P.G., Y.W. and K.M.; resources, J.D., N.L. and Z.Z.; data curation, X.H.; writing—original draft preparation, X.H.; writing—review and editing, J.P. and Y.L.; visualization, X.H.; supervision, Y.H.; project administration, J.P.; funding acquisition, J.P. and Y.L. All authors have read and agreed to the published version of the manuscript.

**Funding:** This work was supported by Matching Funds of Natural Science Foundation of Tianjin (Grant No. 19JCZDJC33700 to J.P.); National Key R&D Program of China (Grant No. 2020YFC1521800 to Z.Z. and J.D.); The National Natural Science Foundation of China (Grant No. 82073832 to Y.L.); Tianjin Natural Science Fund for Distinguished Young Scholars (Grant No. 20JCJQJC00070 to Y.L.); Tianjin Municipal Education Commission Scientific Research Project (Natural Science, Grant No. 2019ZD11 to Y.L.).

**Institutional Review Board Statement:** Not applicable.

**Informed Consent Statement:** Not applicable.

**Data Availability Statement:** Not applicable.

**Acknowledgments:** We gratefully acknowledge the assistance of Dawa Shen from Chinese Academy of Cultural Heritage and Guanglan Xi from National Center of Archaeology.

**Conflicts of Interest:** The authors declare no conflict of interest.

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
