# Peer review of "Inhibitory Effect of Cinnamaldehyde on Main Destructive Microorganisms of Nanhai No. 1 Shipwreck"

_applsci, doi:10.3390/app11115262_

Round 1

Reviewer 1 Report

The manuscript "Inhibitory effect of cinnamaldehyde on main disease microorganisms of Nanhai No.1 Shipwreck" (Manuscript ID: applsci-1197413) presents interesting data about the microbial flora present on a Chinese wreck and the damage that these microorganisms can cause. The treatment currently in use to limit the proliferation of these microorganisms is illustrated and, given the toxic effects of the used chemical , the authors attempt to experiment other safer molecules, specifically cinnamaldehyde. 

The work presents interesting and quite understandable data; despite this, there are several issues that need to be solved before the manuscript is published. In particular, there are some recurring or relevant errors throughout the text that absolutely need to be corrected: 

  • throughout the manuscript, Bacillus meqaterilum NK-NH10 is indicated among the various isolated bacteria. Looking in the bibliography indicated (references of the same authors, see reference 8), it turns out how this bacterium is indeed Bacillus megaterium. This is a serious mistake, given the possible misunderstanding with one of the many species belonging to the genus Bacillus. I kindly ask the authors to check the other scientific names entered as well.
  • the various microorganisms tested are indicated throughout the manuscript as "disease microorganisms". Not causing a real disease, I believe that the term "disease" should be changed with words more relevant to the context, such as "damaging", "degrading", "injourius", "harmful", "destructive" and so on.
  • the materials and methods section is presented in some parts in an excessively concise manner, almost as if it were a check list. I kindly ask to the authors to formally review this part without modifying its content, which is considered correct. 

In addition to these changes, other minor revisions listed below are required.

Line 50: change "the special buffer" with "a specific buffer".

Line 54: "Some bacteria and fungi can destroy wood by degrading cellulose": please provide examples with relative references. 

Line 56: change "cultural relics" with a synonym.

Lines 62-63: delete the sentence.

Lines 64-71: broaden the discussion on the toxic effects of Euxyl K100, both in relation to the environment (e.g. this compound is very harmful to fish) and to human health, in order to better support the discussion on the use of harmless compounds. 

Line 67: change "human friendly" with a synonym (e.g. health friendly).

Line 70: change "It is our research goal" with "Our research goal is".

Lines 74-75: please rephrase the sentence. 

Lines 77-80: "Cinnamaldehyde has significant antibacterial effect": in the next sentence, in addition to bacteria, you also speak of fungi. Correctly indicate the effects of cinnamaldehyde, both in relation to bacteria and fungi.

Lines 87-93: indicate how these microorganisms were isolated and, in the case of stored and revitalized strains, how this procedure was performed. 

Line 95: specify what is meant by CMC agar.

Line 113: specify what is meant by LB medium.

Line 121 and later: how was cinnamaldehyde obtained? Please, specify.

Line 140: specify what is meant by PDA agar.

Line 153: change "0.003125-2 mg/mL and Euxyl® K100 at 0.003906-5 mg/m" with "0.003125-2 mg/mL and Euxyl® K100 at 0.003906-5 mg/mL".

Line 164: change "parallel" with "replicates".

Line 189: primers must be indicated in uppercase (Eub338, Eub518). In this first mention of the primers, it would be useful to include their sequence as well. 

Line 194: specify the taq polymerase/commercial mix used, indicating the manufacturer. 

Line 209: "showed higher degradation ability": compared to what? Specify or rephrase the sentence. 

Line 232: "Comparable to the effect of 0.5% Euxyl® K100 inhibition". Instead, looking at Figure 2 you can see how it depends on the cinnamaldehyde dosage. Please, specify. 

Lines 257-259: move this sentence into discussions section.

Line 326: delete Euxyl. 

Line 345: change "chip" with "disk".

Line 355: change "alter the structure of cell membranes" with "alter its structure".

Figure S1: change "Microbial growth on the dichotomy plate; left (bacteria)" with "Microbial growth on the dichotomy plate; left (microorganism)".

Table S1: similarity is indicated, but not with respect to which specific microorganism. Do you mean the percentage of similarity towards a reference strain? Specify this information, possibly introducing the GenBank Accession Number of the sequences of the microorganisms with which the match was obtained. 

Reviewer 2 Report

This study is technically sound, however I have some remarks regarding the introduction:

L17: Southern Song Dynasty, China,

L65-66: The authors need to provide a reference for the harmful effect on the environment and humans.

L76-77: “For example, gum, ice cream and other foods contain cinnamaldehyde” It is better to delete this sentence. If the authors would like to keep, then please add examples for medicine and other cosmetics.

L77: “Cinnamaldehyde has significant antibacterial effect” this is redundant.

L87: delete “disease”, or it can be changed to “pathogenic”.

L88-93: This part is not clear, did the authors isolate these microorganisms? Or these are just the target organisms. Please rewrite this part.

Round 2

Reviewer 1 Report

The authors replied to the requested comments in an exhaustive manner. Despite this, some minor adjustments remain to be made before the manuscript publication.

Line 53: change "reproduction" with "replication".

Line 55: abbreviate the genus of Trichoderma reesei (T. reesei) as already indicated; change "Bacteroides succinogenes" with "Fibrobacter succinogenes" (change in taxonomic classification).

Line 91: change "Naihai" with "Nanhai".

Line 95: change "purified" with "identified".

Line 332: change "microbial growth" with "microbial replication".